# GCTD3: Modeling of Bipedal Locomotion by Combination of TD3 Algorithms and Graph Convolutional Network

**Khoi Phan Bui** [1,*][ID]**, Giang Nguyen Truong** [1][ID] **and Dat Nguyen Ngoc** [2][ID]

1   School of Mechanical Engineering, Hanoi University of Science and Technology, Hanoi 100000, Vietnam; trgiang071098@gmail.com
2   Department of Computer, Control and Management Engineering, Sapienza University of Rome, 00185 Roma, Italy; datnguyen.roma@gmail.com
*   Correspondence: khoi.phanbui@hust.edu.vn; Tel.: +84-913-525160

**Abstract:** In recent years, there has been a lot of research using reinforcement learning algorithms to train 2-legged robots to move, but there are still many challenges. The authors propose the GCTD3 method, which takes the idea of using Graph Convolutional Networks to represent the kinematic link features of the robot, and combines this with the Twin-Delayed Deep Deterministic Policy Gradient algorithm to train the robot to move. Graph Convolutional Networks are very effective in graph-structured problems such as the connection of the joints of the human-like robots. The GCTD3 method shows better results on the motion trajectories of the bipedal robot joints compared with other reinforcement learning algorithms such as Twin-Delayed Deep Deterministic Policy Gradient, Deep Deterministic Policy Gradient and Soft Actor Critic. This research is implemented on a 2-legged robot model with six independent joint coordinates through the Robot Operating System and Gazebo simulator.

**Keywords:** GCTD3; GCN; TD3; ROS; reward function; bipedal robot

## 1. Introduction

This article aims to improve the reinforcement learning (RL) algorithm in discovering the locomotion of the bipedal robot by adding some kinematic constraints to the algorithm. The GCTD3 algorithm using Graph Convolutional Network (GCN) [1,2] exploits the robot's graph-like structure to enhance the Twin-Delayed Deep Deterministic Policy Gradient (TD3) algorithm in the design locomotion of a bipedal robot. In addition, thanks to the neural network input aggregated from previous states and a reward function built, based on two human walking states, the balance of the robot body is improved significantly during the robot's movement.

Reinforcement Learning is one of the most favored methods for recent robot learning fields; it results in robots that are autonomous and flexible in performing several specific tasks. Although the application in practice of reinforcement learning algorithms to real-world robots is still difficult and challenging [3], there are still some applications that are implemented in practice, especially in the field of robotics [4–9]. In the recent decade, reinforcement learning has been applied to leg-robots in a variety of ways [10–14]. For example, the model-free method [12] does not use or simplify a dynamic model and also does not use information about the kinematics and center of mass of the robot. With the physics-based locomotion problem [14], a model was learned based on a real physical model, where the human's walking and kicking skills are processed, stored and adapted in the robot's locomotion. The model-based methods [11,13] were applied when the robot's configuration was known, or the robot was modeled to find the joint movements based on the calculation of the value function. The disadvantages of the methods are the large amount of computation and were more attentive to the physical design of the robot than on the kinematic parameters and constraints between the joints.

Moreover, the traditional kinematic methods [15–17] were based on the kinematic relationship between the joints, so there are some difficulties related to the motion trajectory of the robot. The methods using the Denavit–Hartenberg matrix [15] and the Jacobi matrix [16,17] represent the relative coordinates of the coordinate systems lying on the joints, as well as their derivatives. The advantage of these methods is that they are easy to implement, but are not flexible, with regular changes in gait in accordance with the multifunctional actions of the robot. This means that the robot's trajectory is always changing. Meanwhile, the robot's walking is controlled on the basis of the developed motion trajectory, as well as the dynamic model of the robot. In work Ref. [18], a fuzzy logic-based controller has been applied, eliminating the difficulties in calculating robot dynamics. However, the control is still based on the pre-designed motion trajectory.

In our method, the robot's walking is trained in two states, including (1) one leg standing, one foot swinging, and (2) two feet touching the ground to switch roles. The robot's movement is based on such a change of state in accordance with the human gait. The robot learns this through our built-in rewards function.

The actor-critic method [19] has many algorithms that are widely applied to deal with the above problems such as [20–23]. The general idea of using this method is to learn a policy by using the policy function and the value function simultaneously. In this research, an extension of the TD3 [20] algorithm was proposed to include more information about the connection between the joints of the robot in the training process. In fact, there are many articles [20–25] using reinforcement learning algorithms such as TD3, DDPG and SAC to find the desired angle values of the joints of the robot. However, their algorithms only used the information about the velocity and angular value of the joints for training, they did not take advantage of the graph topology and the binding relationship of the humanoid robot, as in our method. Our method is compared with algorithms [20–22] in Section 4, which shows the high efficiency of this method.

The authors were inspired by the fact that the robot's joints have geometric relations, for example, the knee joint is related to the thigh joint or the foot joint is related to the knee joint, so that this paper uses GCN to represent the constraints of robot joints above and integrate GCN into the reinforcement learning algorithm. In practice, Graph Neural Network (GNN) often applies to problems in the field of chemistry [26,27] that require chemical bonding between the molecules, or GNN is often used to recommend web-scale [28]. The relationship between the joints of a bipedal robot has a similarity to the relationship between elements in chemistry. In several problems related to human joints, GCN achieved high efficiency when applied to predict actions in a video after extracting points on the human body into the form of a skeleton [29–32]. In the field of reinforcement learning, GCN has been applied to the multi-agent problem where it finds out the relationship between agents for cooperative tasks [33]. There has been no research using graph neural networks for the robot walking problem before. The authors found that taking advantage of the features obtained from GCN before putting these features into the reinforcement learning algorithm gave better results than previous reinforcement learning algorithms.

This article contributes a method dealing with the robot locomotion problems for the bipedal robot. Besides combining GCN with the TD3 algorithm to supplement the linking information of the joints, this article also builds a suitable reward function to achieve high efficiency. A two-legged robot model with six independent joint coordinates was also built by the authors for the implementation process. The process of training the model as well as testing the results is implemented through the Robot Operating System (ROS), which is suitable and has many utilities for robot control [34–36]. Simulation is performed in the Gazebo simulator [37,38] that can be easily combined with ROS. Simulation results obtained after training are recorded in this link: https://youtu.be/t1MVmvCIWr0 (accessed on 30 January 2022) (Supplementary Material).

The content of this paper includes five sections: introduction, network architecture, training process, evaluation and conclusion. The introduction section gives ideas and proposes the GCTD3 method; the network architecture section aims to present in detail

the theoretical basis of GCTD3 algorithm; the training process section depicts how to implement the algorithm in ROS and Gazebo as well as our experience in training process; the evaluation section is to compare our proposed method with other RL methods through evaluation metrics and result graphs; and the last section presents the conclusions drawn from this research.

## 2. Network Architecture

### 2.1. Graph Convolutional Network

GCN is a convolutional neural network and is based on graph theory, the main components of a graph neural network are Node (N) and Edge (E) [1]; the nodes will be connected to each other through directed edges or undirected edges in order to represent the relationship and influence of the nodes to each other in the graph.

Figure 1 is an example of a graph; this graph depicts the influence of nodes on each other. For example, all three nodes, D, C and A, act on each other, E and D are two nodes that have no direct relationship between them.

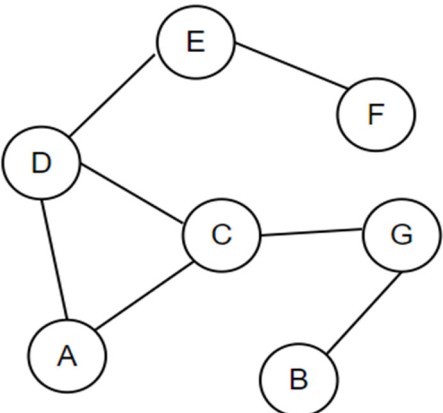

**Figure 1.** Nodes and edges in a graph structure.

The relationship between nodes is represented by an adjacency matrix $A_{N \times N}$, where N is the number of nodes in the graph and the elements of the adjacency matrix A are set according to the principle: $A_{ij} = 1$ when the $i^{th}$ node and $j^{th}$ node are connected and vice versa if there is no edge between the $i^{th}$ and $j^{th}$ node, $A_{ij} = 0$ (with $i, j \in [1, N]$). According to the above principle, the adjacency matrix of the graph of Figure 1 is represented as the Equation (1).

$$A_{7 \times 7} = \begin{array}{c} \\ A \\ B \\ C \\ D \\ E \\ F \\ G \end{array} \overset{\begin{array}{ccccccc} A & B & C & D & E & F & G \end{array}}{\begin{bmatrix} 0 & 0 & 1 & 1 & 0 & 0 & 0 \\ 0 & 0 & 0 & 0 & 0 & 0 & 1 \\ 1 & 0 & 0 & 1 & 0 & 0 & 1 \\ 1 & 0 & 1 & 0 & 1 & 0 & 0 \\ 0 & 0 & 0 & 1 & 0 & 1 & 0 \\ 0 & 0 & 0 & 0 & 1 & 0 & 0 \\ 0 & 1 & 1 & 0 & 0 & 0 & 0 \end{bmatrix}} \tag{1}$$

After obtaining the matrix depicting graph A, the matrix A is normalized to the form of a Laplacian matrix $\Delta_{norm} = I_N - D^{-1/2} A D^{-1/2}$ [39] for training with the neural network using the gradient-based method. Where $I_N$ is an identity matrix of size $N \times N$, and $D$ is a diagonal matrix with the elements on the principal diagonal equal to the degrees of the nodes $D_{ii} = \sum_j A_{ij} (1 \leq i, j \leq N)$. Since $\Delta_{norm}$ is a symmetric positive definite matrix, $\Delta_{norm}$ can be decomposed into $\Delta_{norm} = I_N - D^{-1/2} A D^{-1/2} = U \Lambda U^T$. $U$ and $\Lambda$ are matrices representing eigenvectors and eigenvalues of $\Delta_{norm}$, respectively.

An input signal $x \in \mathbb{R}^N$ is filtered by $g_\theta = diag(\theta)$ (with $\theta \in \mathbb{R}^N$) which is defined in (2):

$$g_\theta \star x = U g_\theta(\Lambda) U^T x \tag{2}$$

In this Equation (2), the function $g_\theta(\Lambda)$ (3) is approximated by Chebyshev polynomial according to Equation (4) [1,39].

$$g_\theta(\Lambda) \approx \sum_{k=0}^{K} \theta_k T_k\left(\widetilde{\Lambda}_{norm}\right) \tag{3}$$

$$with \begin{cases} T_0(x) = \mathbf{1} \\ T_1(x) = x \\ T_k(x) = \mathbf{2}x T_{k-1}(x) - T_{k-2}(x) \end{cases} \tag{4}$$

In (3), $\widetilde{\Lambda}_{norm} = \frac{2}{\lambda_{max}}\Lambda - I_N$ is the matrix $\lambda$ rescaled to the maximum eigenvalue $\lambda_{max}$ and $\theta_k \in \mathbb{R}$ is the Chebyshev coefficient of degree $k$. From (2), (3) and $\widetilde{\Delta}_{norm} = \frac{2}{\lambda_{max}}\Delta_{norm} - I_N$, the convolution of signal $x$ is rewritten as (5).

$$g_\theta \star x \approx \sum_{k=0}^{K} \theta_k T_k\left(\widetilde{\Delta}_{norm}\right) x \tag{5}$$

To facilitate the training process of GCN and reduce the number of learning parameters, $k$ is chosen to be 1, $\widetilde{\Delta}_{norm} \approx \Delta_{norm} - I_N$ and $\theta = \theta_0 = -\theta_1$, so the Equation (5) is rewritten as (6):

$$\begin{aligned} g_\theta \star x &\approx \theta_0 T_0\left(\widetilde{\Delta}_{norm}\right) x + \theta_1 T_1\left(\widetilde{\Delta}_{norm}\right) x \\ &= \theta_0 x + \theta_1 \widetilde{\Delta}_{norm} x \\ &= \theta_0 x + \theta_1 (\Delta_{norm} - I_N) x \\ &= \theta(I_N - \Delta_{norm} + I_N) x \\ &= \theta\left(I_N + D^{-1/2} A D^{-1/2}\right) x \end{aligned} \tag{6}$$

According to the paper [23], a renormalization trick was used for (6) to avoid exploding and vanishing gradient problems: $I_N + D^{-1/2} A D^{-1/2} \to \hat{D}^{-1/2} \hat{A} \hat{D}^{-1/2}$ (with $\hat{A} = A + I_N$ and $\hat{D}$ is a diagonal degree matrix of $\hat{A}$. In general, input $X \in \mathbb{R}^{N \times F}$ is a feature matrix of N nodes, each node has F features, and we obtain an output matrix $Y \in \mathbb{R}^{N \times F'}$ (7). It can be seen that each input node has F features, after being multiplied by the matrix $\Theta \in \mathbb{R}^{F \times F'}$, it will be characterized by a F'-dimensional vector. The matrix $Y$ is calculated as (7):

$$Y = \hat{D}^{-1/2} \hat{A} \hat{D}^{-1/2} X \Theta \tag{7}$$

### 2.2. Twin Delayed Deep Deterministic Policy Gradient

The implementation of reinforcement learning algorithms such as DDPG, SAC and TD3 all follow the Markov Decision Process (MDP) as sequences (*S*, *A*, *S'*, *R*), where *S*, *A*, *S'* and *R* are state, agent, next state and reward, respectively. The trained agent gets two elements (a state containing information from the environment and a reward calculated based on the previous state) to find suitable actions, and the agent will then perform these actions in the next state. The loop will continue as the environment sends the next state and rewards to the agent to find the next action. Each set of parameters (*S*, *A*, *S'*, *R*) received will be stored in a replay buffer *B* to serve the training process of the model.

The main purpose of reinforcement learning is to find an optimal policy $\pi_{opt}$ (with $\pi : S \to A$) for the highest cumulative reward [40,41], the cumulative reward is represented as (8):

$$R_t = \sum_{i=t}^{T} \gamma^{i-t} r(s_i, \ a_i) \tag{8}$$

where $\gamma$ is a discount factor ($0 < \gamma < 1$), $R_t$ is the total reward the agent receives from timestep t to the end of an episode and the reward at each timestep is $r(s_i, a_i)$.

For problems that need to compute continuous actions through each state, the actor-critic method is appropriate. The actor network receives an input state that the agent obtained from the environment and calculates the actions, and the actor network is the policy that the agent needs to find an optimal one. The critic network returns a $Q$-value that is used to evaluate the calculated actions from the actor network, and during the training process, the learning parameters are updated to maximize this $Q$-value.

TD3 is one of these actor-critic methods, the structure and the steps of TD3 algorithms [20] are presented as follows:

- The algorithm uses an actor network $\pi_\phi(s)$, two critic networks $Q_{\theta_1}(s,a)$, $Q_{\theta_2}(s,a)$, corresponding to these three networks are their target networks: one target actor network $\pi_{\phi'}(s')$ and two target critic networks $Q_{\theta_i'}(s', a')$ ($i = 1, 2$). Where $\phi$ and $\theta_i(i = 1, 2)$ are the learning parameters of the actor and critic network, respectively, and similarly $\phi'$ and $\theta_i'$ are the parameters of the target actor network and the target critic networks;

- In order for the agent to explore a variety of states in the environment, actions computed from the actor network have a noise added $\epsilon$ [42]. This noise helps the data in replay buffer $B$ to be augmented, and the noise in TD3 algorithms follows a Gaussian distribution [20]. Both the action "$a$" and the target action "$a'$" are added a noise $a \sim \pi_\phi(s) + \epsilon$, $a' \sim \pi_{\phi'}(s') + \epsilon$ (with $\epsilon \sim \mathcal{N}(0, \sigma)$). The Bellman equation [43] is used to calculate a target value $y(r, s')$ as (9). In Equation (9), the smaller value between two outputs ($Q_{\theta_1'}$, $Q_{\theta_2'}$) of target critic networks is fed into the Bellman equation to avoid overestimating the $Q$-value:

$$y(r, s') = r + \gamma \min_{i=1,2} Q_{\theta_i'}(s', \pi_\phi(s') + \varepsilon); \tag{9}$$

- The parameter sets ($\theta_1$ and $\theta_2$) are updated by minimizing the loss values $L_{\theta_1, B}$ (10) and $L_{\theta_2, B}$ (11) which are the expected value of the difference between the target value $y(r, s')$ and the two $Q$-value, where $(s, a, r, s')$ are retrieved from replay buffer $B$:

$$L_{\theta_1, B} = \mathop{\mathbb{E}}_{(s,a,r,s') \sim B}\left[\left(Q_{\theta_1}(s, a) - y(r, s')\right)^2\right] \tag{10}$$

$$L_{\theta_2, B} = \mathop{\mathbb{E}}_{(s,a,r,s') \sim B}\left[\left(Q_{\theta_2}(s, a) - y(r, s')\right)^2\right] \tag{11}$$

The parameter set $\phi$ of the actor network is updated by maximizing $Q_{\theta_1}$ value $\max_\phi \mathop{\mathbb{E}}_{s \sim B}\left[Q_{\theta_1}(s, \pi_\phi(s))\right]$;

The parameters of the target networks are not updated consecutively like those of the actor and critic networks in order to avoid overestimation during the training process. These parameters are updated after a certain number of timesteps (in this research, the delay timestep is 2), in addition, a hyperparameter $\tau$ ($0 < \tau < 1$) is used to make updating these weights slower. These parameters of the target networks are updated as the Equations (12) and (13):

$$\theta_i' \leftarrow \tau\theta_i + (1 - \tau)\theta_i' \tag{12}$$

$$\phi' \leftarrow \tau\phi + (1 - \tau)\phi' \tag{13}$$

*2.3. GCTD3*

In the paper [20], the actor and critic networks used contiguous Fully Connected (FC) layers, so the weights learned only from these FC layers made the model more difficult to achieve convergence. Thus, the paper [9] proposed a method of combining RNN into TD3 to increase input information from many previous consecutive states, however that made the neural network more complex and difficult to train with weak hardware. In this paper, we propose to use the GCTD3 network for two purposes:

- The graph convolutional layers of our GCTD3 show efficiency with data in the form of the graph structure of the bipedal robot. The joints will learn each other's relationships and constraints through these layers, the joints that are not connected will not affect each other. For example, the right knee joint and the right hip joint can share their attributes with each other, but the left knee joint and the right knee joint are not directly related;
- GCNs do not need to use a large number of weights to increase the joint features of the robot joints, so the computational volume is not too large even when using many previous states combined as the input of the neural network. According to Equation (7), the number of weights $\Theta$ used for a joint to increase the number of its features from $F$ to $F'$ is only equal to $F \times F'$.

In this paper, we modelled a 2-legged robot with six independent joint coordinates to test the algorithm (details on the robot model are presented in Section 3.1). We created a graph with seven nodes located at the joints of the robot and one node located at the ground, thereby building the adjacency matrix for the bipedal robot. The order of the robot's nodes is shown in Figure 2, the adjacency matrix $\hat{A}$ and the diagonal degree matrix $\hat{D}$ built according to the principle in Section 2.1 are also shown in Figure 2.

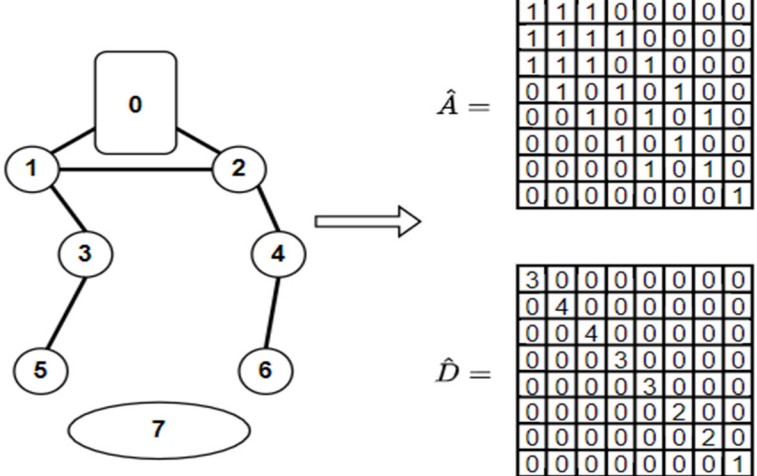

**Figure 2.** The order of nodes, linking edges for the bipedal robot; the adjacency matrix $\hat{A}$ and the diagonal degree matrix $\hat{D}$ built based on the defined nodes and edges.

The characteristics of a node in the next state will be calculated based on the characteristics of that node in the previous state and the neighboring nodes affecting that node. This relationship is modeled in the form of Graph Convolutional layers and based on Equation (7), the layers of graph convolution network are represented as (14):

$$H^{l+1} = \sigma\left(\hat{D}^{-\frac{1}{2}}\hat{A}\hat{D}^{-\frac{1}{2}}H^{l}W^{l}\right) \qquad (14)$$

where $H^{l+1}$, $H^{l}$ are the features matrices of the nodes at $(l+1)^{\text{th}}$ and $l^{\text{th}}$ layers, $W^{l}$ is the weights' matrix at the $l^{\text{th}}$ layer trained by the neural network and $\sigma()$ is the activation function, in this paper we used ReLu function.

The neural network structure used is shown in Figure 3. The input of the algorithm is the sensor signals placed at eight positions consisting of the joints of the robot, the body of the robot, and the ground. In order to increase the input information for the neural network, the most recent five consecutive states are used (His). At a state, each node is characterized by a 2-D input feature vector (In) consisting of the position on the z-axis and the velocity on the y-axis of the body robot node, angular velocity and the angular position of the joints of the robot and the ground contact states of two feet. Since the last five states are used, each node will have a total of 10 features, so the input dimension of the GCTD3

is (8, 10). After three layers of the GCN with the number of hidden features 32, 32 and 16, respectively, each node will be represented by a 16-features vector, these features are the relationships between the nodes affecting each other. Then, we concatenate the features of eight nodes before introducing two fully connected layers and a last linear layer to give the desired angles for the joints of the robot for the next state.

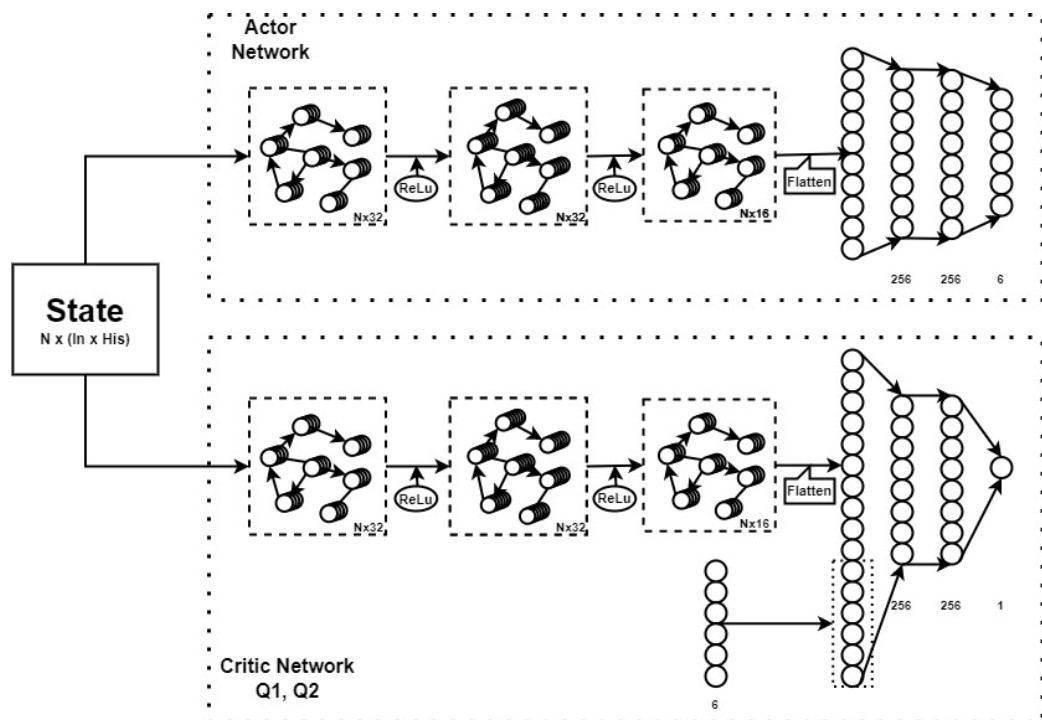

**Figure 3.** The actor network and two critic networks structure in GCTD3 algorithm: N is the number of nodes in the graph; "In" is the number of features of each node at a state; and "His" is the number of most recent consecutive states used during training, the activation function is ReLu.

### 3. Training Process

*3.1. Simulation Environment*

An overview of the robot training and simulation process is shown in Figure 4. ROS is used to transmit signals between the Gazebo simulation environment and the RL algorithm. The model receives the state from Gazebo when the robot moves in the environment, then calculates the reward corresponding to that state to put the input state into the GCTD3 algorithm and find the appropriate action for the robot in the next state. Nodes and topics communicate with each other thanks to ROS which is shown in Figure 5. In Figure 5, when the robot walks in the Gazebo simulation environment, /gazebo node publishes data about the robot's transformation and the ground contact state of the feet to topics /Joint_states, /FootR_contact and /FootL_contact. After that, the /walker_controller node subscribes data provided from the three topics so that RL algorithms can process and find a proper action. These found values are published to six topics corresponding to six robot joints and /gazebo node continues getting these six values from the six topics for the bipedal robot to perform in the next state. The components used in testing the algorithm are detailed in this section.

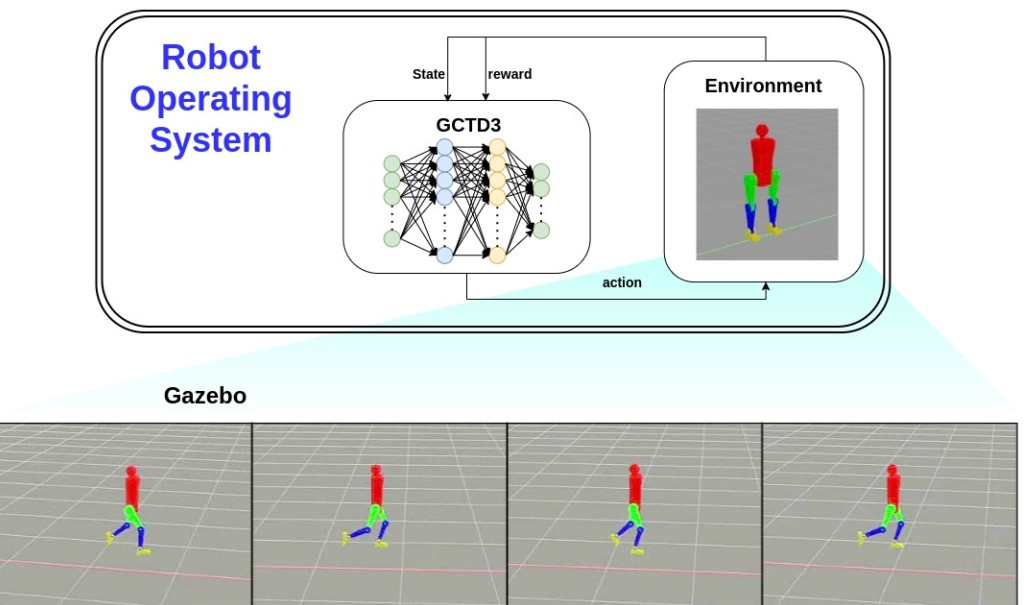

**Figure 4.** Diagram of the main components in the training and simulation process of the bipedal robot.

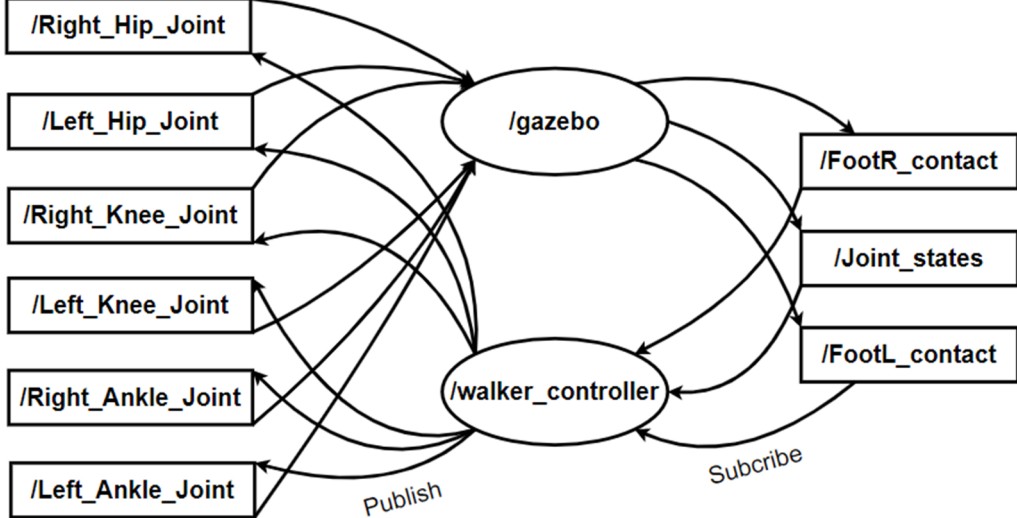

**Figure 5.** Communication between nodes and topics in ROS. Ovals represent nodes and rectangles represent topics.

The robot model has the shape and size as shown in Figure 6. The bipedal robot consists of seven links (the left and right thigh, the left and right shank, the left and right foot and a body), we selected Acrylonitrile butadiene styrene (ABS) material for all links of the robot. The weight and height of each link are presented in Table 1.

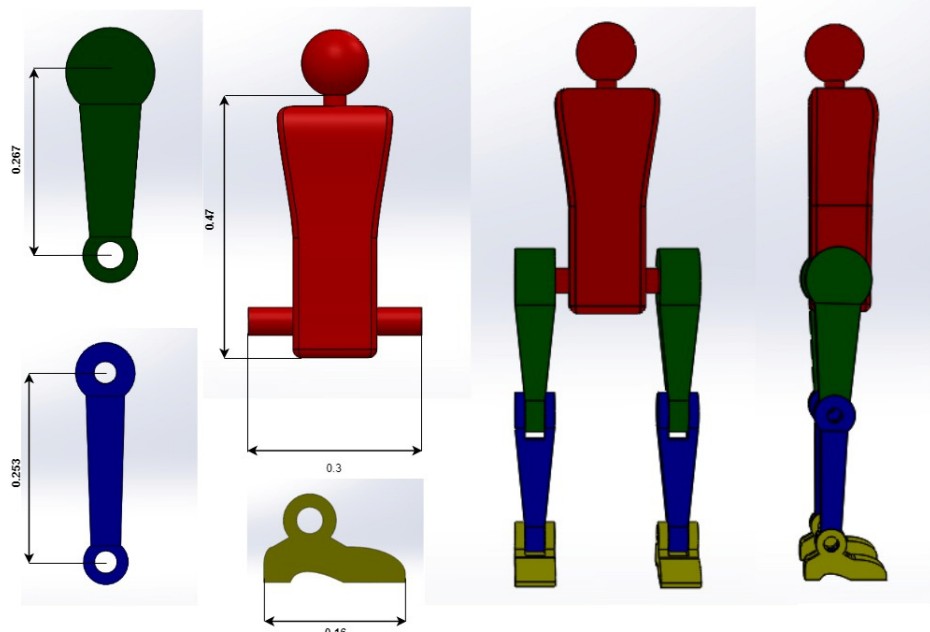

**Figure 6.** Shape and size of the bipedal robot's links.

**Table 1.** Mass, height and material of links of robot.

| Link | Material | Mass (kg) | Height (m) |
|------|----------|-----------|------------|
| Body | Acrylonitrile butadiene styrene (ABS) | 4.0200 | 0.47 |
| Left Thigh | Acrylonitrile butadiene styrene (ABS) | 1.7324 | 0.267 |
| Right Thigh | Acrylonitrile butadiene styrene (ABS) | 1.7324 | 0.267 |
| Left Shank | Acrylonitrile butadiene styrene (ABS) | 0.7527 | 0.253 |
| Right Shank | Acrylonitrile butadiene styrene (ABS) | 0.7527 | 0.253 |
| Left Foot | Acrylonitrile butadiene styrene (ABS) | 0.4111 | 0.065 |
| Right Foot | Acrylonitrile butadiene styrene (ABS) | 0.4111 | 0.065 |

The software used for the simulation is Gazebo and the Robot Operating System is used to get state information from the environment for computational processing to find the appropriate action and then send it back to the environment.

- Environment: Gazebo simulation environment receives control signals through the ROS with a frequency of 50 Hz, i.e., ROS will take the state from the environment and calculate the reward of that state to store data into a replay buffer every 0.02 s;
- State: At each joint of the robot model, sensors are placed to measure the information received from the environment for the agent to process. The sent signals contain information of the angular position and angular velocity of the joints, the coordinates of the robot's body in the vertical direction and the robot's speed according to the robot's movement direction, and the signals from the sensor located at the feet to know whether the robot touches the ground or not, five consecutive states from $t-5$ to $t$ are concatenated again to form the input state of the neural network during training;
- Action: Before sending the robot control signals, the action corresponding to the desired angle coordinates of the joints is calculated by our proposed network, the action consists of six values corresponding to the desired angle of six joints: left hip joint, right hip joint, left knee joint, right knee joint, left ankle joint and right ankle joint. The angle ranges of these joints are limited as shown in Table 2;
- Reward: We use a dense reward function based on the actual human gait to make the algorithm easier to converge. While walking, the robot always has two states: one-foot contacts with the ground and two feet contact with the ground (Figure 7). In each state, the height of the body robot has different values. In the paper [25], only an average

value of the body height during motion is used as a basis height for the robot to learn, two average values of the body height corresponding to two grounding states of the legs in motion are used in this paper. At the single phase of walking (Figure 7a,b), the average height of the robot's body reaches a higher value than that at the double phase of walking (Figure 7c,d). In addition, the robot will also be punished if it performs actions that delay its movement such as falling or standstill, the details of the reward function are presented in Table 3.

**Table 2.** Limiting rotation angle and annotation of joints.

| Joint | Flexion (Rad) | Extension (Rad) | Annotation |
|---|---|---|---|
| Hip | 0.7854 | 0.7854 | $\alpha$ |
| Knee | 1.3962 | 0.0012 | $\beta$ |
| Ankle | 0.7854 | 0.7854 | $\gamma$ |

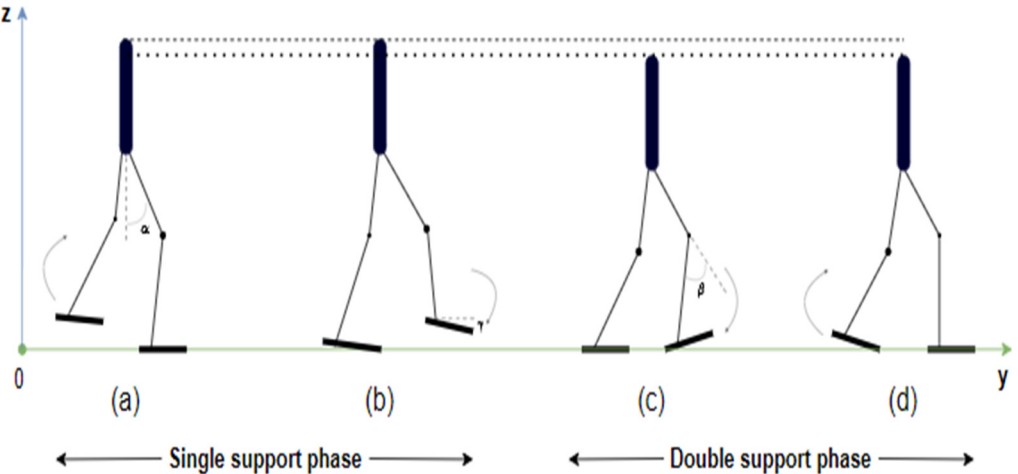

**Figure 7.** The stage of the bipedal robot's walk: (**a**,**b**) are two single phases of walking; (**c**,**d**) are two double phases of walking.

**Table 3.** Bonus and punishment values at state t of the reward function during robot motion.

| Situation | Reward |
|---|---|
| Bonus | $r_i = \min(v_{iy}, v_{\text{target}}) + 0.2 \cdot (y_i - y_{i-h})$ |
| Gait punishment | $r_i{-} = c_1 \lvert z_i - z_{\text{target}} \rvert$ <br> $with \begin{cases} z_{\text{target}} = -0.01 & : \text{Single support phase} \\ z_{\text{target}} = -0.025 & : \text{Double support phase} \end{cases}$ |
| Long ground contact time | $r_i{-} = 1.0$ |
| Fall down | $r_i{-} = 10.0$ |

### 3.2. Experiment

The algorithm is trained for more than 10,000 epochs, in each epoch, the robot starts to move from the starting position to the end of the distance in 15 m or falls. The other hyperparameters of the actor and critic networks of the GCTD3 algorithm chosen to train the model are presented in Table 4.

**Table 4.** Hyperparameters values of Actor and Critic networks used for the training process.

|  | **Actor** | **Critics (Q1, Q2)** |
|---|---|---|
| Episode | | 10,000 |
| Policy update frequency | | 2 |
| Learning rate | $3 \times 10^{-4}$ | $3 \times 10^{-4}$ |
| Weight decay $\tau$ | 0.001 | 0.001 |
| Optimizer | Adam | Adam |
| Hidden GC layers | [32, 32, 16] | [32, 32, 16] |
| Hidden fully connected layers | [256, 256] | [256, 256] |
| Discount factor | 0.99 | 0.99 |

After training the RL algorithms, the rewards obtained during the training process are shown in Figure 8. In Figure 8, the dashed lines are the total reward values of the timesteps at each episode, the solid line is the average cumulative reward of the episodes. According to the graph in Figure 8, we can see that after more than 10,000 training episodes, the cumulative reward of an episode using GCTD3 (the solid red line) is higher than the rest of the algorithms.

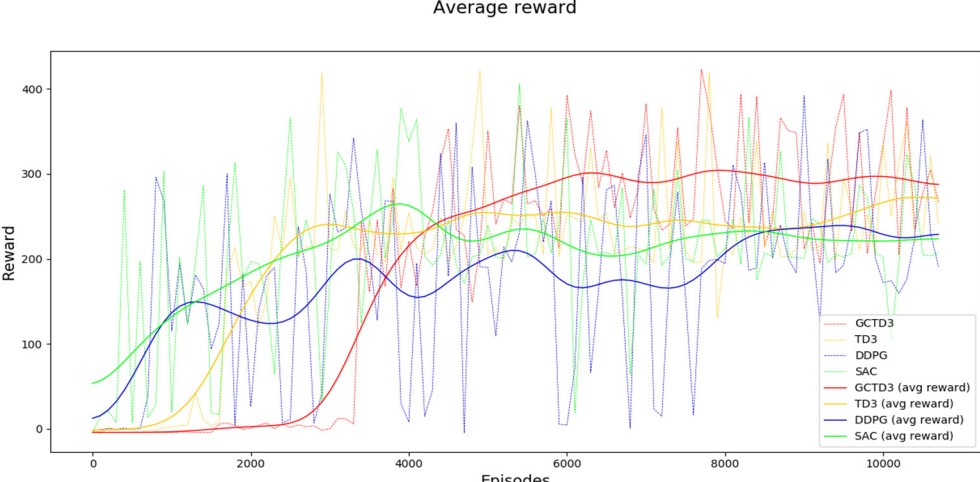

**Figure 8.** Cumulative reward over 10,000 episodes when training GCTD3 (red), TD3 (yellow), DDPG (green) and SAC (blue).

## 4. Evaluation

In this section, we will show some evaluation results between our GCTD3 algorithm and the TD3 baseline and we will also make a comparison with other RL algorithms. Figure 9 shows that the period and amplitude of oscillation of the knee and ankle joints when using GCTD3 are more stable than TD3 over timestep. In addition, when comparing these characteristics between the left and right legs of the robot, the GCTD3 algorithm also achieved better similarity of the trajectory of the two legs than TD3.

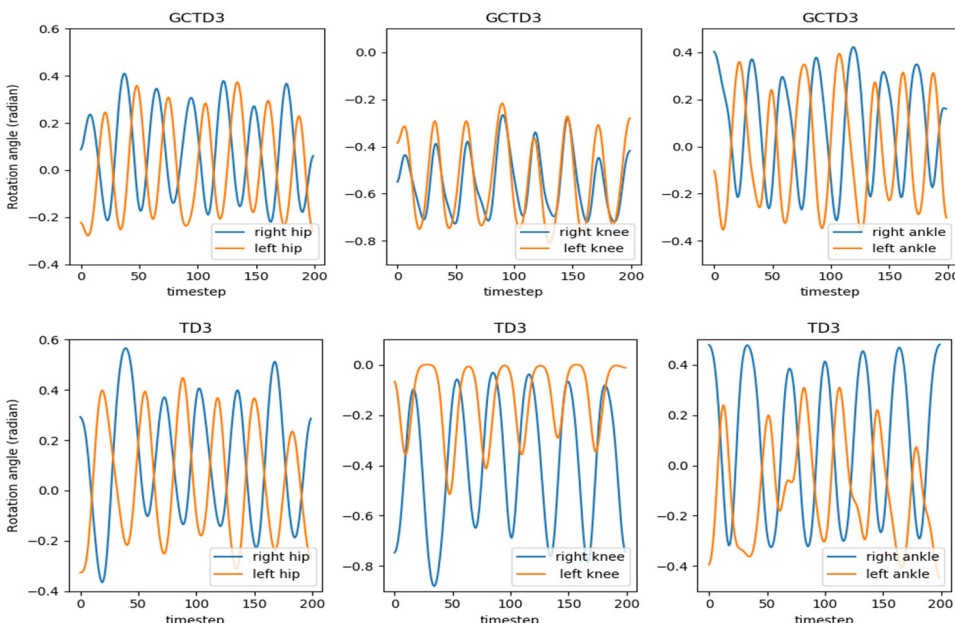

**Figure 9.** Graph of rotation angle values of hip, knee and ankle joints (from left to right) of GCTD3 and TD3 algorithms (from top to bottom). The blue and orange lines represent the rotation of the joints of the right and left legs over timestep, respectively.

In addition, according to the graph of Figure 10, we can see that the body position when using the GCTD3 algorithm fluctuates more stably and cyclically than the other three algorithms. In Table 5, the robot's cumulative reward value of GCTD3 is also higher than that of other algorithms, the robot also has a moving speed of 1.414 (m/s) that is faster than DDPG (1.344 m/s), SAC (0.619 m/s) and TD3 (1.164 m/s).

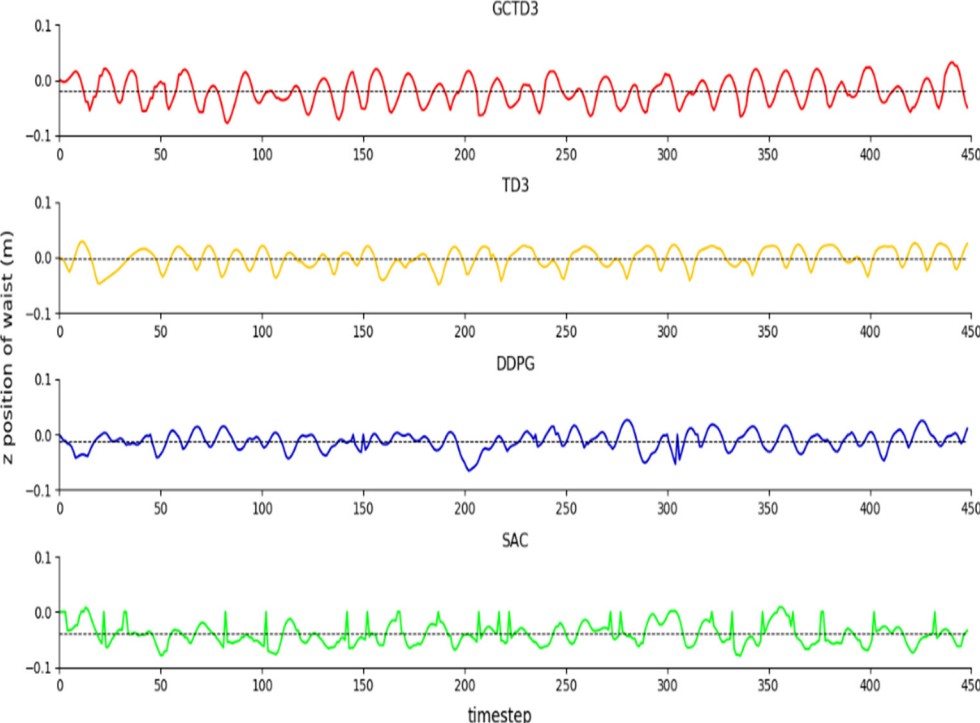

**Figure 10.** Graph of the position of robot's body oscillation during walking when using reinforcement learning algorithms GCTD3, TD3, DDPG, and SAC.

**Table 5.** Average rewards, mean body position, and average robot velocity obtained when evaluating algorithms in 50 episodes.

|  | DDPG | SAC | TD3 | GCTD3(our) |
|---|---|---|---|---|
| Average reward | $241.28 \pm 89.44$ | $224.18 \pm 56.96$ | $370.31 \pm 37.26$ | **$380.0 \pm 18.47$** |
| Position of the body robot (m) | $-0.0114$ | $-0.0397$ | $-0.0020$ | $-0.0202$ |
| Velocity (m/s) | $1.344$ | $0.619$ | $1.164$ | **$1.414$** |

## 5. Conclusions

This paper presents the GCTD3 method using graph convolutional layers in the TD3 baseline algorithm to apply to the bipedal robot having six independent joint coordinates. The GCTD3 algorithm achieves a higher average cumulative reward and a higher average speed than the TD3 algorithm as well as other reinforcement learning algorithms (DDPG and SAC) which is shown in Table 5. Our method exploits the graph structure of the robot's joints to help the robot move more smoothly, and it helps the rotation angles of the joints to have a better cycle (Figure 9). In this paper, through the actual observation of human gait, a reward function of RL algorithms was built, based on two phases (single phase of walking and double phase of walking). The results evaluated and performed through the ROS and Gazebo environment proved the effectiveness of our method and it is a premise to apply this method in a real environment with real robots in future studies. Moreover, the GCTD3 can be improved to orient the robot to perform other more complex actions such as jumping, avoiding obstacles and moving on complex terrains, and it can be extended to apply for more multi-legged and degrees-of-freedom robots such as quadrupeds or spider robots.

**Supplementary Materials:** The following supporting information can be downloaded at: https://www.mdpi.com/article/10.3390/app12062948/s1.

**Author Contributions:** Conceptualization, K.P.B., G.N.T. and D.N.N.; methodology, K.P.B., G.N.T. and D.N.N.; software, G.N.T.; validation, K.P.B. and D.N.N.; writing-original draft preparation G.N.T. and D.N.N.; writing-review and editing, K.P.B.; visualization K.P.B.; supervision, K.P.B. and D.N.N.; project administration, K.P.B. All authors have read and agreed to the published version of the manuscript.

**Funding:** This research received no external funding.

**Institutional Review Board Statement:** Not applicable.

**Informed Consent Statement:** Not applicable.

**Data Availability Statement:** Not applicable.

**Conflicts of Interest:** The authors declare no conflict of interest.

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
