# Peer review of "GCTD3: Modeling of Bipedal Locomotion by Combination of TD3 Algorithms and Graph Convolutional Network"

_applsci, doi:10.3390/app12062948_

Round 1

Reviewer 1 Report

An interesting article is presented by the authors.

The English used in the article is good, only a minor spellcheck is required.

The number of references is adequate, and they provide the necessary background for the article. Since the reviewer believes that the results are well-presented, his remarks only contain formatting issues.

Citations should be written as, for example, [1, 2] not [1][2].

Line 338: The reviewer believes that a period should be used instead of a comma in the number (1,414 -> 1.414). Also, "that is" is missing before the word "faster".

Figures and Tables should be mentioned in the text before they are placed in the article. The most noticeable Figures in this regard are 4 and 5. There is no text before them.

Subfigures should be referred in the text as, for example, 7 (a), not 7.a

Figure 8: The limits on the y axis should be the same on all plots. That way, the differences could more easily be seen.

In the References section, abbreviated journal names should be used when citing journal articles.

Author Response

Dear reviewer,

We would like to thank you for your valuable comments and suggestions. In the revision, each and every comment or suggestion from you has been carefully considered and revised. The attached file is the summary to describe the revision as well as the response to reviewer’s comments. We also used “Track Changes” function of MS Word to highlight the modification in the revised manuscript.

Reviewer 2 Report

This paper presents a new reinforcement learning algorithm for modeling bipedal locomotion.  The research is interesting.  The proposed algorithm seems reasonable and sound, but I did not check the correctness of the equations.  The paper is well-written.  

I would like to suggest the authors use the past tense to describe the previous work.  For example, in Line 199, "the paper [9] proposes..." should be "the paper [9] proposed ...".

Author Response

(The authors gave the same response as above.)

Reviewer 3 Report

Dear authors

Please see the attached report.

sincerely yours.

Author Response

(The authors gave the same response as above.)
